# Transcriptional Interference Regulates the Evolutionary Development of Speech

**DOI:** 10.3390/genes13071195

**Published:** 2022-07-04

**Authors:** Douglas P. Mortlock, Zhi-Ming Fang, Kelly J. Chandler, Yue Hou, Lissett R. Bickford, Charles E. de Bock, Valsamma Eapen, Raymond A. Clarke

**Affiliations:** 1Department of Molecular Physiology and Biophysics, Center for Human Genetics Research, Vanderbilt University School of Medicine, Nashville, TN 37232-0700, USA; mortlock@chgr.mc.vanderbilt.edu (D.P.M.); chandler.kelly@epamail.epa.gov (K.J.C.); yue.yuehou@gmail.com (Y.H.); bickford@vt.edu (L.R.B.); 2Discipline of Psychiatry, Ingham Institute, University of New South Wales, 1 Campbell Street Liverpool, Sydney, NSW 2170, Australia; z.fang@unswalumni.com (Z.-M.F.); v.eapen@unsw.edu.au (V.E.); 3Children’s Cancer Institute, Lowy Cancer Research Centre, UNSW Sydney, Kensington, NSW 2052, Australia; cdebock@ccia.org.au

**Keywords:** larynx, pisiform, wrist evolution, primate gene, primate evolution, bone morphogenetic protein, transcriptional interference, overlapping gene, gene complex, GDF6

## Abstract

The human capacity to speak is fundamental to our advanced intellectual, technological and social development. Yet so very little is known regarding the evolutionary genetics of speech or its relationship with the broader aspects of evolutionary development in primates. In this study, we describe a large family with evolutionary retrograde development of the larynx and wrist. The family presented with severe speech impairment and incremental retrograde elongations of the pisiform in the wrist that limited wrist rotation from 180° to 90° as in primitive primates. To our surprise, we found that a previously unknown primate-specific gene *TOSPEAK* had been disrupted in the family. *TOSPEAK* emerged de novo in an ancestor of extant primates across a 540 kb region of the genome with a pre-existing highly conserved long-range laryngeal enhancer for a neighbouring bone morphogenetic protein gene *GDF6*. We used transgenic mouse modelling to identify two additional *GDF6* long-range enhancers within *TOSPEAK* that regulate *GDF6* expression in the wrist. Disruption of *TOSPEAK* in the affected family blocked the transcription of *TOSPEAK* across the 3 *GDF6* enhancers in association with a reduction in *GDF6* expression and retrograde development of the larynx and wrist. Furthermore, we describe how *TOSPEAK* developed a human-specific promoter through the expansion of a penta-nucleotide direct repeat that first emerged de novo in the promoter of *TOSPEAK* in gibbon. This repeat subsequently expanded incrementally in higher hominids to form an overlapping series of Sp1/KLF transcription factor consensus binding sites in human that correlated with incremental increases in the promoter strength of *TOSPEAK* with human having the strongest promoter. Our research indicates a dual evolutionary role for the incremental increases in *TOSPEAK* transcriptional interference of *GDF6* enhancers in the incremental evolutionary development of the wrist and larynx in hominids and the human capacity to speak and their retrogression with the reduction of *TOSPEAK* transcription in the affected family.

## 1. Introduction

The breath-taking utility of the human vocal apparatus allows for thoughts and information encoded by language to be communicated through speech. The larynx and tongue are the primary organs of speech. In early infancy the human larynx and tongue have a more superior position as in non-hominoid mammals, from where they gradually descend during postnatal development starting from ~3–4 months of age [1,2,3,4,5,6] (Figure 1). This descent of the larynx frees the thyroid cartilage from its close contact with the hyoid bone thus relieving constraints on the mobility and utility of the larynx that endows the young child with the capacity to speak [1,2,3,4,5,6] (Figure 1). The human larynx descends further during puberty increasing its size and changing its structure (Figure 1). During the prepubescent period, the larynx of males and females are of approximately equal size and the angle of the thyroid cartilage is ~120° in both females and males. During the pubertal period in males, the thyroid cartilage enlarges significantly more than in females and the angle of the thyroid cartilage decreases from ~120° to ~90° which gives rise to the protrusion (prominence) of the thyroid cartilage commonly referred to as ‘Adam’s apple’. This increase in the anteroposterior length of the thyroid cartilage is ~3× greater in males than in females. During this pubertal period of expansion of the thyroid cartilage and the vocal cords, which are attached to the inner surface of the thyroid cartilage, also undergo elongation in males more than females. By comparison, non-hominoid primates including Cercopithecoidea (Old World monkeys) and Ceboidea (New World monkeys) maintain superior (infantile) positioning of the tongue and laryngeal cartilages, with the thyroid cartilage locked in its superior position with the hyoid and where the infantile epiglottis maintains contact with the soft palate [3,4,5,6]. In chimpanzee the infantile epiglottis persists, allowing chimps to eat and breathe simultaneously comparable to the human infant [1,2,3,4,5,6].

Concurrent with the phylogenetic descent and reconfiguration of the larynx in hominoids was another pivotal evolutionary development of the skeleton in hominoids that involved reconfiguration of the radio-ulnar joint of the wrist dramatically increasing the angle of wrist rotation enabling brachiation [7,8]. The evolutionary development of brachiation in hominoids was dependent upon the separation of the carpus, and more especially the pisiform, from the ulnar. This process involved reductions in the size of the styloid process of the ulnar together with changes to the carpus in particular a phylogenetic sequence of reductions in the length of the pisiform bone in hominoids. The pisiform is the smallest of the carpal bones enclosed within the flexor carpi ulnaris tendon of the wrist and is thus classed as a sesamoid bone (Figure 2). The phylogenetic sequence of reductions in the length of the pisiform in hominoids effectuated the incremental retraction of the pisiform away from the styloid process of the ulnar in the hominoids: gibbon, orangutan, chimpanzee, gorilla and human, respectively [7,8]. This in turn facilitated a dramatic increase in the flexibility of the radio-ulnar joint of the wrist increasing its angle of rotation/supination from ~90° to ~180°. This in turn enabled brachiation in what is considered to be one of the defining features in the divergence of the hominoid line [8]. Coincident with this evolutionary remodelling of the pisiform was the remodelling of the other carpals in the wrist in particular the triquetrum and lunate (Figure 2) [7]. All eight carpal bones are cartilaginous at birth and begin to ossify at different rates from within the first 60 days of postnatal development [9]. The first carpal to fully ossify is the hamate, then the triquetrum, lunate, scaphoid, trapezium, and ossification of the trapezoid is complete by ~7 years of age. The pisiform is the last of the carpals to fully ossify by ~12 years of age [9]. Together these findings indicate strong selective pressure for the incremental phylogenetic reductions in the length and size of the pisiform (wrist) and the incremental descent and growth of the larynx during the evolution of the hominoids, hominids and human, respectively [9].

In this study, we provide evidence that the phylogenetic descent and maturation of the larynx during the evolution of speech in hominoids shares a common molecular and mechanistic pathway with that of the reduction and retraction of the pisiform from the radio-ulnar joint during hominoid evolution that involves the emergence and incremental evolution of a primate-specific gene *TOSPEAK* which evolved a human-specific promoter. We report a speech impaired family with disruption of *TOSPEAK* associated with the retrograde descent, growth, morphology and flexibility of the larynx concordant with retrograde increases in the length of the pisiform that severely limit wrist rotation. Affected family members displayed an amazing incremental series of elongations of the pisiform that represent the inverse (retrogression) of those incremental reductions in the length of the pisiform that had occurred during the progressive evolution of the wrist in hominoids [7,8]. Likewise, there were variations in the degree of laryngeal deformation and laryngeal descent in affected family members that were the inverse (retrogression) of the descent and maturation of the larynx that had occurred during hominoid evolution [1,2,3,4,5,6].

The incremental and retrogressive deformation of the larynx and wrist in the affected family segregated with the disruption of a primate-specific non-coding gene that we named *TOSPEAK*/*C8orf37AS1*. In this study, we used a transgenic approach to investigate developmental gene regulation in the genomic region now spanned by *TOSPEAK* in primates. We discovered that the *TOSPEAK* gene had emerged de novo in a relative of extant primates, across a region of the genome with a series of pre-existing long-range enhancers for the *growth and differentiation factor 6* (*GDF6*) gene. *GDF6* encodes a bone morphogenetic protein (BMP) that regulates joint, ligament, tendon, cartilage and bone formation through extracellular signalling at select sites in the larynx and wrist and ankle, the middle ear and spine, and the coronal sutures between the bones of the skull [10,11,12,13].

## 2. Materials and Methods

### 2.1. Radiology and MRI Analyses

Radiology of the wrist was performed using routine methodology [14]. MRI sagittal T1 and T2 and axial T2 weighted sequences of the head and neck were performed using MRI Tesla 3T imaging.

### 2.2. Primate Tissues

We acquired fixed archival opportunistic tissue specimens for Chimpanzee: *Pan troglodytes* also referred to here as Chimp, Gorilla: *Gorilla gorilla gorilla*, Orangutan: *Pongo pygmaeus* also referred to here as Pongo, Gibbon: *Muelleri*, Old world monkeys, Brazza: *Cercopithecus neglectus*, Rhesus: *Macaca mulatta* and Crab eating macaque: *Macaca fascicularis* also referred to here as CE Macaque, New world monkeys, Marmoset: *Callithrix jacchus* and Cotton-top Tamarin: *Saguinus Oedipus*, and Lemur: *Lemur catta* (gifts from the Australian Museum of Natural History, Sydney, Australia). We isolated DNA and RNA from tissues and prepared cDNA using routine methodology. We performed PCR on DNA samples and rtPCR on RNA samples (see below) and sequenced the amplicons before comparing the nucleotide sequences with that of human.

### 2.3. RT-PCR Characterisation of TOSPEAK Transcripts

RT-PCR reactions contained 5 μL of the diluted cDNA template, 2.5 μL of 10× PCR buffer, 0.2 μL of 25 mM dNTPs, 1 μL of each of the forward and reverse primer stocks (10 mM) (12), 1.5 μL of 25 mM MgCl_2_ and 0.25 μL of AmpliTaq Gold polymerase (Thermofisher Scientific, Sydney, Australia) made up to 25 μL with ddH_2_O and amplified using an initial denaturation step at 94 °C for 10 min followed by 40 cycles at 94 °C for 30 s, 58 °C for 30 s and 72 °C for 40 s and a final extension of 72 °C for 15 min.

### 2.4. Comparative Genome Analyses

Nucleotide sequences from a ~900 kb region of the genome 3′ of the *GDF6* gene locus were extracted from the Ensembl and NCBI GenBank databases (http://www.ensembl.org/biomart/martview accessed on 10 June 2022, version 41.36c; http://www.ncbi.nlm.nih.gov/genbank/ accessed on 10 June 2022, Build 37.1) for human, chimpanzee, dog, mouse, and opossum and analysed for any evolutionary conservation using VISTA (http://genome.lbl.gov/vista/ accessed on 10 June 2022) [15] using the human sequence as the reference sequence and applying strict selection criteria for highly conserved regions (HCRs ≥ 200 bp ungapped alignment with >90% identity).

### 2.5. Luciferase Assays

Luciferase assays performed in mammalian cells transfected with pGL3-p-*TOSPEAK* constructs containing primate promoter sequences (+5 to −254 bp relative to human sequence) from *TOSPEAK C8ORF37AS1[+9]* and compared with either 1 ug of the parent pGL3-basic vector or with the *Firefly luciferase* reporter gene. Expression constructs were cotransfected with 10 ng of pHRG-B Renilla luciferase control plasmid to normalise transfection efficiency as described elsewhere [16]. Reporter activity was measured using the dual luciferase assay system (Promega Coorporation, Alexandria NSW Australia) with values expressed as mean fold increase of Firefly luciferase/Renilla luciferase ± standard deviation and normalised to the pGL3-basic empty vector control [16].

### 2.6. BAC Transgenic Analyses

A region spanning approximately −150/+110 Kb surrounding the mouse *Gdf6* gene has been previously described [17]. In this study, 5 BAC clones spanning the ~900 kb genomic region 3′ of *Gdf6* were chosen for analysis using the UCSC genome browser. The addresses and predicted insert end coordinates (based on BAC end sequences and July 2007 assembly, NCBI37/mm9) of these clones were, with increasing distance from *Gdf6*: RP23-115F19 (chr4: 9,832,336–10,039,639 RP23-444O1 (chr4: 10,000,431–10,163,998); RP23-56O6 (chr4: 10,157,134–10,395,586); RP23-333E9 (chr4: 10,329,803–10,537,032); RP23-354G12 (chr4: 10,532,029–10,720,157).

Briefly, BAC DNAs were used for reporter assays in transgenic mouse embryos by either co-injecting embryos with wild-type BACs with a *Hsp68* promoter-*LacZ* reporter plasmid, or with BACs that had been engineered to carry the *Hsp68-LacZ* cassette in the pBACe3.6 backbone. Pronuclear co-injection of BACs and free plasmid was carried out as described previously [18]. Engineering BACs was completed as follows: A *Hsp68* promoter-*LacZ* cassette from *pSfi-HspLacZ* [19] was linked to an FRT-flanked tetracycline resistance cassette. BAC recombineering was used to insert the fragment into the pBACe3.6 vector backbone of the BAC clones, using methods and tetracycline selection as previously described [19]. The insertion was internal to the SacB gene and located between bases 3002–3003 of pBACe3.6. The modified BACs, or BAC + plasmid mixtures were injected into mouse embryos, and the embryos were collected at E14.5 for staining with X-gal. Images were captured on an Olympus stereomicroscope with DP70 digital camera. Results from modified BAC clones were similar to the co-injection method (see Section 3).

## 3. Results

### 3.1. Speech Impairment Associated with Evolutionary Retrograde Laryngeal Configuration

We describe a family with speech impairment and limited wrist rotation and grasping due to evolutionary retrograde development of the larynx and wrist. An earlier study found this family to have reduced expression of the *GDF6 bone morphogenetic protein gene* which had been shown to have a role in larynx and wrist development in the mouse [10,11,12]. Varying degrees of speech impairment in this family were found to be associated with the deformation of the thyroid cartilage of the larynx [12,13,14,20]. The pisiform in the wrist was also deformed and there was a discrete pattern of ossification/fusion of select joints of the carpals, tarsals, ossicles and vertebrae (Figure 3) [12,13,14,20]. The degree of speech impairment in affected family members varied dramatically. The more severe cases of speech impairment were males that presented with an aphonic whisper that was totally ineffectual with background noise. There were a range of milder cases of speech impairment, that were mostly female’s some of which displayed a faint breathy whisper and others a weak husky voice and one affected male member of the family with the least degree of vertebral fusion had no obvious speech impairment [12,13,14]. Spectroscopic analysis of the larynx indicated that the degree of speech impairment correlated with the degree of deformation of the larynx.

In this study, we used MRI to better understand deformation of the larynx in the affected family. MRI of the head and neck of a male member of the family with speech impairment from a young age (III-6* Figure 3) identified superior (evolutionary retrograde) displacement of the hyoid and thyroid cartilages with a shortened tongue, small mouth and overcrowding of the teeth (Figure 4A,B) [3,4,5,6]. MRI further indicated aberrant ossification of the thyroid-hyoid ligament extending between the greater cornu of the hyoid and the thyroid cartilage that appeared to have retarded the descent of the thyroid cartilage during puberty which had failed to form the ‘prominence’ of the thyroid cartilage due to a failure of the thyroid cartilage to tilt and protrude as expected during male puberty. Spectroscopic analysis found the vocal cords, which attach to the inner surface of the prominence of the thyroid cartilage, were shorter (Figure 4C,D) [20] and of a whiter colour as opposed to the normal pink colour suggesting a change in composition and possibly a degree of ossification (Figure 4C,D) [20]. Together these findings indicated aberrant ossification and stiffening of laryngeal ligaments and tendons that restricted both the normal postnatal descent of the larynx and the morphological expansion and angulation of the thyroid cartilage during male puberty [1,2,3,4,5,6]. In addition, there were discrete patterns of postnatal ossification of the ossicles of the middle ear in association with age-related conductive hearing loss in females from an early age [13]. The developmental descent of the external ear was also impeded in some affected family members and the skull was misshapen possibly due to the fusion of coronal sutures. Together these findings indicated retarded and retrogressive postnatal positioning of pharyngeal and laryngeal elements in the affected family together with infantile tongue, mouth and mandible [12,13].

### 3.2. Restricted Wrist Rotation Associated with Evolutionary Retrograde Wrist Morphology

In addition to retrograde descent of the larynx, the speech affected family presented with retrograde morphology of the pisiform in the wrist (Figure 4E–K). We found the pisiform was expanded in size to varying degrees in all affected family members tested (Figure 4E–K). Expansion of the pisiform resulted in its elongation proximally and its intrusion into the radio-ulnar joint (Figure 5) [7,8,12] comparable to the retrograde configuration found in non-hominoid primates (Figure 4E–K). Elongation of the pisiform was bilaterally symmetrical within affected family members [12] yet varied greatly in length between affected family members (Figure 4E–K). In those affected family members with maximum elongation of the pisiform it extended deeper into the radio-ulnar joint where it restricted supination of the wrist from ~180° to ~90° as is the case found in non-hominoid primates (see Introduction for review) [7,8]. In some family members the pisiform formed a novel articulation with the styloid process of the ulnar (Figure 5) [8]. Furthermore, when the pisiforms from all of the affected family members were aligned it was evident that they displayed a series of incremental elongations (Figure 4E–K) which was the inverse of the incremental shortening of the pisiform that had occurred during hominid evolution that had facilitated the retraction of the pisiform away from the radio-ulnar joint thus increasing its flexibility (supination) from ~90° to ~180° in hominoids. In more severely affected family members, there was also distal extension of the pisiform leading to hamate-pisiform coalition (Figure 4K and Figure 5). In some affected family members, there was also restricted apposition of the thumbs that restricted grasping and writing. A discrete fusion was identified between the lunate and triquetrum in the carpals of some affected family members (Figure 5) and between select tarsals [13].

### 3.3. Primate-Specific Gene TOSPEAK Disrupted in Speech Impaired Family

Our earlier studies of the affected family had found that speech impairment segregated with the disruption of a long non-coding gene which we named *TOSPEAK* [12]. In the present study, we performed comparative DNA sequence analysis of *TOSPEAK* between species. To our surprise we found that *TOSPEAK* was a primate-specific gene that had evolved rapidly in hominoids (Figure 6A). *TOSPEAK* was present in all primates tested yet absent from the genomes of all non-primates tested including bat, dog, mouse and rat (Table 1). Moreover, there was no evidence of any *TOSPEAK* exons in any of the non-primates tested. Together these findings indicated that *TOSPEAK* had emerged de novo in an ancestor of extant primates with all of the essential elements of an active transcription unit (Table 1). *TOSPEAK* had retained these essential elements of transcription in all primates tested. Furthermore, *TOSPEAK* was transcribed in all primates tested (Table 1). Albeit DNA and RNA comparative sequence analysis indicated that the exon sequence and structure of *TOSPEAK* were not conserved in the primate lineage (Figure 6 and Table 1) [22]. In hominids there had been an increase in the number of *TOSPEAK* exons (Table 1) and the polyadenylation signal now used in human had only recently emerged de novo in hominids (Figure 6B and Table 1). Moreover, only the first and last exons of *TOSPEAK* (exons 1 and 9, respectively) were permanently spliced into all transcripts. All transcripts were small (≤600 nucleotides) and poorly conserved in sequence with a high incidence of stop codons in all reading frames of all transcripts (Table 1). We raised antibodies against a potential short open reading frame identified in the most abundantly expressed transcript and found no evidence of a *TOSPEAK* protein [12]. Together these findings indicated *TOSPEAK* to be a primate-specific non-coding gene that had not been conserved in exon sequence or structure in primates and that it was alternatively spliced into a wide range of short poorly conserved transcripts rich in stop codons.

### 3.4. TOSPEAK Proximal Core Promoter and Start Site Perfectly Conserved

The core promoter of the *TOSPEAK* gene was found to have a perfectly conserved proximal region (+2 to −18) and a highly variable distal region (−19 to −85) (Figure 7A). The proximal region of the core promoter which adjoins the transcription start site (−18 to +2) was found to be perfectly conserved in all primates tested (Figure 7A). This was consistent with the conservation of *TOSPEAK* transcription in all primates tested (Table 1). Moreover, we found no other conserved 20 nucleotide string in any of the many *TOSPEAK* transcripts (Table 1). This proximal region of the core promoter therefore represented the longest conserved nucleotide string anywhere within the *TOSPEAK* transcripts and core promoter regions combined (Figure 7A and Table 1). This in turn indicated conserved functionality of the proximal core promoter and conservation of *TOSPEAK* transcription which was consistent with our finding that *TOSPEAK* was transcribed in all primates tested (Table 1).

### 3.5. Incremental Increase in TOSPEAK Promoter Strength in Higher Hominids with Human Having the Strongest Promoter

In contrast to the conservation of the proximal region of the core promoter of *TOSPEAK*, the adjoining more distal region of the core promoter (−19 to −85) was found to be a hot spot of sequence variation in hominoids (Figure 7A). Alignment of the core promoters for the different primate species (−19 to −120) indicated many base changes during primate evolution including the de novo emergence of a penta-nucleotide ‘CGGGG’ element in the core promoter of the crab eating macaque (Old World Monkey: between −41 to −45 from the transcription start site of *TOSPEAK*; Figure 7A). This ‘CGGGG’ element was subsequently found to be duplicated in gibbon to form a ‘CGGGGCGGGG’ direct tandem repeat (between −41 to −50). This direct repeat was conserved in orangutan and chimpanzee and subsequently expanded to form 3 direct tandem repeats in gorilla and between 7–11 direct tandem repeats in human (between −19 to −85 nucleotides from the transcription start site of *TOSPEAK*; Figure 7A).

We compared the strength of these primate promoter variants by engineering the promoters of the different primate species into reporter gene constructs (Figure 7B). In vitro assays found that the strength of the *TOSPEAK* promoter had increased incrementally in direct correlation with the emergence and incremental expansion of the ‘CGGGGCGGGGC’ tandem direct repeat within the core promoter of the cotton-top tamarin (no ‘CGGGG’ element), chimpanzee (1 tandem direct repeat), gorilla (3 tandem direct repeats) and human (7 tandem direct repeats), respectively (Figure 7). The human promoter had the highest number of direct tandem repeats and drove the highest level of transcription in all cell lines tested (Figure 7B).

### 3.6. TOSPEAK Is Transcribed across GDF6 Long-Range Enhancers

Given that *TOSPEAK* transcription was conserved in primates and incrementally increased in hominids we searched for features of *TOSPEAK* that might account for positive selective pressure on *TOSPEAK* transcription in primates and hominids. We subsequently found that *TOSPEAK* harboured a highly conserved *GDF6* long-range enhancer (ECR5) within one of its introns [12]. Moreover, ECR5 was found to regulate *GDF6* transcription in the developing pharyngeal arches [23], which in primates give rise to the laryngeal structures deformed in the speech impaired family (Figure 4B,D). As the family had also presented with retrograde expansion of the pisiform and fusion of carpals and tarsals we searched for additional *GDF6* enhancers in that genomic region 3′ of *GDF6* now spanned by *TOSPEAK* in primates. We developed a series of transgenic mouse models containing contiguous regions of the mouse genome 3′ of the *GDF6* gene including across the entire region now spanned by *TOSPEAK* in primates. These regions of the genome were cloned upstream of a reporter gene in bacterial artificial chromosome (BAC) constructs (Table 2). Mouse embryonic stem cells were subsequently transfected with these BAC constructs to generate transgenic mice. Interrogation of reporter gene expression during development in this series of transgenic mice identified two additional *GDF6* enhancers within the genomic region spanned by *TOSPEAK* (Figure 8). These two enhancers which we named DJE1 and DJE2 were distal joint enhancers located one on either side of the highly conserved pharyngeal/laryngeal *GDF6* enhancer ECR5 within *TOSPEAK* (Figure 6 and Figure 8). DJE1 regulated *GDF6* transcription in the developing pisiform (arrowed, Figure 8ii). Both DJE1 and DJE2 positively regulated *GDF6* transcription within the developing distal limb joints of the carpals and tarsals (Figure 8ii). In this respect, all three *GDF6* enhancers located within *TOSPEAK* (in order DJE1, ECR5 and DJE2—see Figure 8) regulated tissue specific expression of *GDF6* during development in tissues that corresponded precisely with skeletal tissues and structures deformed in the speech impaired family wherein *TOSPEAK* was disrupted. This correspondence included the retrograde descent/configuration and deformation of the laryngeal cartilages (ECR5—Figure 8ii and Figure 4B,D) [1,2,3,4,5,6], retrograde elongation of the pisiform (DJE1—Figure 8ii and Figure 4E–K) and remodelling and fusion of the distal limb joints of the carpals and tarsals (DJE1 and DJE2—Figure 8ii,iii and Figure 5)) [14]. No other *GDF6* enhancers were identified in or near the region of the genome spanned by *TOSPEAK* in primates (Table 2) and no *GDF6* enhancers were either disrupted or relocated by the chromosomal breakpoint in the affected family. Notwithstanding, *TOSPEAK* transcription across the *GDF6* enhancers had been blocked by the breakpoint in the affected family (Figure 6) and *GDF6* expression was reduced (Figure 9A) [12,13].

## 4. Discussion

In this study, we describe a family with severe speech impairment and restricted wrist rotation due to evolutionary retrograde development of the larynx and wrist. We report the affected family harboured a breakpoint in a primate-specific gene *TOSPEAK* that blocked its transcription across long-range enhancers for the neighbouring *GDF6 bone morphogenetic protein gene* which regulate the transcription of *GDF6* in the developing larynx and wrist [12].

To date, little is known regarding the precise molecular and biological basis of hominoid and human divergence from the other primates. The genetic difference between human and chimpanzee has been calculated to be ~1.2% (and ~1.6% between human and gorilla) when DNA base changes between shared genes are considered and ~5% when all sequence changes and rearrangements (insertions, deletions and duplications) have been factored into the calculation [24]. However, it is an ongoing endeavour to determine exactly which genes and which DNA changes regulated the evolutionary development of the hominoids. Numerous primate-specific and human-specific genes have been identified. For example, the human-specific gene *ARHGAP11B* which emerged in human by gene duplication has been implicated in the increased size of the human brain. In addition, there are over 59,000 genomic loci which harbour candidate gene regulatory sequences (HSRS) unique to human which were either inherited from extant common ancestors or emerged de novo in the human genome [24]. One relatively common type of HSRS with proven capacity to drive transcriptional change are the short tandem repeats (STRs) located within the core promoters of human genes (within −120 bases from the transcription start site of the gene) 14% of which are human-specific and 28% primate-specific [24,25]. Notwithstanding, the majority of the STRs within these human core promoters have been evolutionarily conserved [24,25,26] where proximity of the STR to the transcription start site of the gene correlates with higher STR conservation scores [26]. The most common STR sequence found in 62% of the core promoters of human protein coding genes is the ‘GC rich’penta-nucleotide repeat ‘CGGGG/GCCCC’ [24] which is the same sequence as the STR that emerged de novo in the core promoter of *TOSPEAK*. A ‘CGGGG’ penta-nucleotide sequence emerged de novo in the core promoter of *TOSPEAK* in Old World monkey or one of its near ancestors (Figure 7A). This ‘CGGGG’ sequence was subsequently duplicated to form a tandem direct repeat ‘CGGGGCGGGG’ in the core promoter of gibbon. This tandem repeat was conserved in orangu-tan and chimpanzee and subsequently expanded to form 3 direct tandem repeats in gorilla before expanding further to form between 7–11 direct tandem repeats in human. This (CGGGGCGGGGC)_7–11_ direct repeat in the core promoter of *TOSPEAK* represents the largest STR of the same sequence found in the core promoters of any of the human protein coding genes. Moreover, the (CGGGGCGGGGC)_7–11_ direct repeat in the core promoter of *TOSPEAK* in human represents an overlapping series of perfect consensus binding sites for the Sp1/KLF family of transcription factors (transcriptional activators and repressors) [24] each of which differs in its tissue expression profile and/or its ability to interact with co-activators and co-repressors of transcription of hundreds of human genes [27]. Furthermore, we found direct correspondence between the number of tandem repeats and Sp1/KLF transcription factor consensus binding sites and the strength of the *TOSPEAK* promoter. Human had the strongest *TOSPEAK* promoter in all cell lines tested (Figure 7) [25].

The de novo emergence and evolution of the core promoter of *TOSPEAK* was found to be a tale of two parts with one part highly variable (discussed above) and the other part perfectly conserved in all of the primates tested. With regard to the latter, that part of the proximal core promoter adjoining the transcription start site of *TOSPEAK* (+2 to −18) was perfectly conserved between all of the primates tested and as such was the only fully conserved 20 nucleotide string found in the entire promoter and mature transcript(s) of *TOSPEAK* combined. The exclusive conservation of only this 20-nucleotide string indicated evolutionary pressure to maintain *TOSPEAK* transcription. Indeed, we found *TOSPEAK* transcription was conserved in all primates tested. In contrast, the transcripts of *TOSPEAK* were not conserved between primates having been derived from a primate-specific long non-coding transcription unit (lncRNA gene) with weak conservation of exon structure and sequence. Moreover, *TOSPEAK* transcripts were all short and enriched with stop codons with no evidence of a protein coding domain. Therefore, in contrast to the perfect conservation of the transcription start site and proximal core promoter of *TOSPEAK*, and the conservation of *TOSPEAK* transcription in primates, the short and poorly conserved transcripts of *TOSPEAK* were not conserved in any meaningful way and were therefore judged highly unlikely to have an important role in the evolution of the primates. This conservation of *TOSPEAK* transcription but not the sequence or structure of the *TOSPEAK* transcripts ultimately led us to question regarding the function of *TOSPEAK* transcription in primates and how this might be related to the reduction in *GDF6* expression in the speech impaired family?

The discrete tissue-specific patterns of expression of *BMP* genes like *GDF6* during development are known to be regulated by tissue-specific enhancers and that these patterns of expression correlate with BMP regulation of skeletal morphogenesis [11,17,18,19,23]. Some of these *BMP* gene enhancers have been located up to 700 kb from the *BMP* genes they regulate [11,17,18,19,23]. Using a transgenic gene expression approach, we discovered *GDF6* enhancers in that region between 400 kb–700 kb 3′ of *GDF6* which was within the same region where *TOSPEAK* had emerged de novo in an ancestor of extant primates. Furthermore, the familial breakpoint in *TOSPEAK* had not disrupted these enhancers but rather had blocked the transcription of *TOSPEAK* across the enhancers (Figure 6). Moreover, the *GDF6* enhancers within *TOSPEAK* were found to regulate *GDF6* transcription in precisely the same skeletal elements affected in the family including the pisiform and the joints of the carpals and tarsals and in the larynx and other midline pharyngeal derivatives including the mandible, mouth and tongue. In the affected family the transcription of *TOSPEAK* across the three *GDF6* enhancers had been blocked (Figure 6) and the level of *GDF6* expression reduced suggesting a role for *TOSPEAK* in regulating *GDF6* transcription at sites regulated by the *GDF6* enhancers in *TOSPEAK* (Figure 9A). Earlier in vitro studies had likewise found supporting evidence that *TOSPEAK* transcription positively regulates *GDF6* transcription (Figure 9B) [12]. Together these findings indicated that *TOSPEAK* transcription across these *GDF6* enhancers interferes with and positively regulates *GDF6* enhancer function and *GDF6* transcription.

*GDF6* is known to have a distinct dose dependent inhibitory effect on ossification and mineralization in later-stage, differentiated chondrocytes and osteoblasts in vitro [28,29]. This dose-dependent ossification effect of *GDF6* was also evident in vivo in the affected family [13] and in *GDF6* knockout mice both of which displayed ossification and bony fusion of carpal and tarsal joints [10,13]. Notwithstanding, a broader ossification phenotype was evident within the affected family compared to the *GDF6* knockout mice [13]. The affected family phenotype included the ossification, stiffening and/or shortening of ligaments and tendons including the hyoid thyroid ligament and the vocal cords of the larynx, the Achilles tendon in the ankle and expansion of the pisiform bone located within the flexor carpi ulnaris tendon of the wrist [10,12,13]. More importantly, this broader *GDF6* phenotype of the affected family closely matched the developmental expression pattern of *GDF6* and the tissue specificity of the *GDF6* long-range enhancers now under the transcriptional influence of *TOSPEAK* in primates (Figure 6). Furthermore, the incremental increases in *TOSPEAK* promoter strength in higher hominoids, corresponded precisely with the incremental evolution of these same structures in hominoids. Moreover, the block on *TOSPEAK* transcription across these enhancers in the affected family corresponded with the incremental retrogression of these same structures. Together these findings provide strong support for the incremental increases in *TOSPEAK* promoter strength in higher hominoids as a positive modulator of *GDF6* enhancer function and *GDF6* transcription in the incremental evolutionary development of the larynx and wrist and the capacity to speak and brachiate in hominoids, respectively. In this respect, the capacity for speech represents a quantum competitive advantage for humans. However, the increased flexibility and improved utility of the radio-ulnar joint in hominoids may have provided greater competitive advantage for brachiation with a stronger more versatile grasp, from an earlier stage of hominoid evolution [1,2,3,4,5,6], while the laryngeal capacity to speak, which continued to develop in higher hominids, may represent a spandrel—of later great effect in human.

## 5. Conclusions

We found precise correspondence between the genotype and phenotype of the speech impaired family. The knockdown of *TOSPEAK* transcription across *GDF6* enhancers, accompanied a knockdown of *GDF6* transcription, and corresponded precisely with the increases in ossification of structures that corresponded precisely with the tissue specificity of the *GDF6* enhancers within *TOSPEAK* including increased ossification of the

Pisiform and its elongation (retrogression) and extension into the radio-ulnar joint that restricted wrist rotation.Hyoid-thyroid ligament of larynx retarded (retrogressed) laryngeal descent causing speech impairment.Vocal cords that reduced their elasticity and retarded their pubertal elongation and the progressive protrusion/prominence of the thyroid cartilage that increased the severity of speech impairment in the affected family (Figure 4C,D).

In addition, there was close reciprocal correspondence between the genotype and phenotype of the speech impaired family and the genotypic and phenotypic evolution of the larynx and wrist in primates. The block on *TOSPEAK* transcription across *GDF6* enhancers and the retrogressive configurations of the larynx and wrist in the affected family represented the inverse of the

Emergence of *TOSPEAK* transcription in primates across these *GDF6* enhancersIncremental increases in *TOSPEAK* promoter strength in hominoidsIncreased descent and expansion of the larynx and the incremental retraction of the pisiform.

However, the precise mechanism for the transcriptional enhancement of the *GDF6* enhancers by *TOSPEAK* has yet to be elucidated. In this respect, investigations into the *B-globin* gene locus may be helpful. Fraser and colleagues found that transcription across the *B-globin* gene enhancer facilitated the secondment of the enhancer to the transcription factories thereby increasing enhancer coupling with, and increased transcription of, the *B-globin* gene [30,31]. This is consistent with the findings in this study regarding *TOSPEAK* transcription across the *GDF6* enhancers (Figure 10) [32].

In summary, we propose the following timeline in the molecular and structural evolution of the capacity for speech.

Conserved long-range tissue-specific enhancers regulate the expression of the ancient *GDF6* gene in its role in regulating the morphology, elasticity and structure of the ancient larynx and wrist joints.The de novo birth of the *TOSPEAK* gene in primates across pre-existing *GDF6* enhancers (DJE1, ECR5 and DJE2) for laryngeal and wrist development (Figure 6).Perfect conservation in primates of the○Proximal core promoter of *TOSPEAK* (Figure 5).○Transcription start site of *TOSPEAK* (Figure 7A).○Transcription of *TOSPEAK* (Table 1).Incremental increases in *TOSPEAK* promoter strength in hominoids and hominids with the strongest *TOSPEAK* promoter in human (Figure 7B) drives incremental increases in the transcription of *TOSPEAK* across *GDF6* enhancers in higher hominoids.Phylogenetic increases in transcriptional interference of site-specific *GDF6* enhancers positively upregulates the transcription of *GDF6* at those sites during development [12].Upregulation of GDF6 resulted in a down-regulation of chondrogenesis (cartilage formation) and endochondral ossification at those sites that in turn facilitated an increase in the flexibility of the ligaments and tendons of the larynx and wrist (Figure 4) [28,29].Increased flexibility of the hyoid thyroid ligament permitted phylogenetic descent of the larynx thus removing hyoid cartilage constraints on the flexibility and utility of the larynx and the tongue in hominids (Figure 4).Increased flexibility of the larynx and tongue increased their utility and the capacity to speak.

The evolution of the overlapping *TOSPEAK/GDF6* gene complex coupled the incremental molecular and structural evolution of both the larynx and wrist with the capacity for speech and wrist rotation (brachiation), respectively (Figure 4) [1,2,3,4,5,6]. The evolution of the capacity to speak represented a quantum competitive advantage for humans. However, the increased flexibility and improved utility of the radio-ulnar joint in hominoids may have provided a greater competitive advantage through brachiation and stronger grasping from an earlier stage of hominoid evolution [1,2,3,4,5,6] while the laryngeal capacity to speak, which continued to develop in hominids, may represent a spandrel—of later great effect. Conversely, the incremental and progressive evolutionary development of the capacity to speak retrogressed with the disruption of *TOSPEAK* in the affected family. This effectively blocked *TOSPEAK* transcription across the site specific *GDF6* enhances (DJE1, ECR5 and DJE2) which ultimately led to a reduction in *GDF6* expression at those sites which in turn caused an increase in the ossification and reduced flexibility of those structures regulated by the *GDF6* enhancers. This in turn retarded the postnatal descent of the thyroid cartilage which reduced the flexibility and utility of the larynx causing severe speech impairment in the affected family (Figure 4).

The findings of this study indicate an important role for *TOSPEAK* transcriptional interference in modulating *GDF6* enhancer function and *GDF6* transcription in the evolutionary development of the larynx and wrist and the structural capacity to speak. Notwithstanding, for this capacity to speak to be fully realized the structural evolution of the larynx must have been accompanied at some stage by an integrated neurological development for speech. In this respect, those autism genes that regulate the neuronal aspects of speech development and control should be considered including those that encode the neurexin trans-synaptic connexus [33]. Interestingly, the *LRRTM3* gene associated with autism and Tourette syndrome which encodes a neurexin 1 ligand, is discordantly regulated by transcriptional interference (https://www.westernsydney.edu.au/newscentre/news_centre/story_archive/2013/uws_researcher_cracks_the_genetic_code_for_tourette_syndrome accessed on June 2022) [33,34,35,36,37].

## Figures and Tables

**Figure 1 genes-13-01195-f001:**
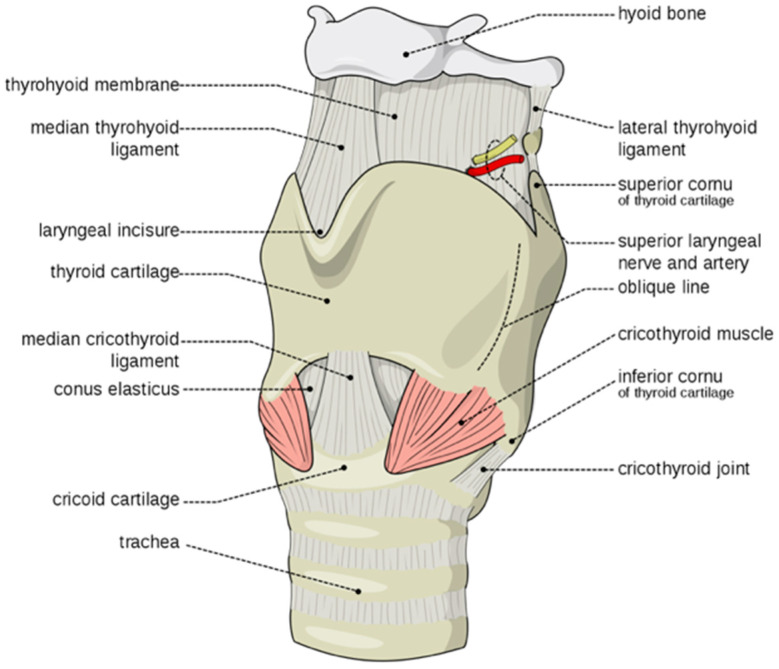
**Laryngeal Development.** Postnatal descent of the human larynx. Human larynx after its postnatal descent separating the thyroid cartilage from the hyoid by Olek Remesz based on: Gray951.png, CC BY-SA 2.5, https://commons.wikimedia.org/w/index.php?curid=3492701 (accessed on 10 June 2022).

**Figure 2 genes-13-01195-f002:**
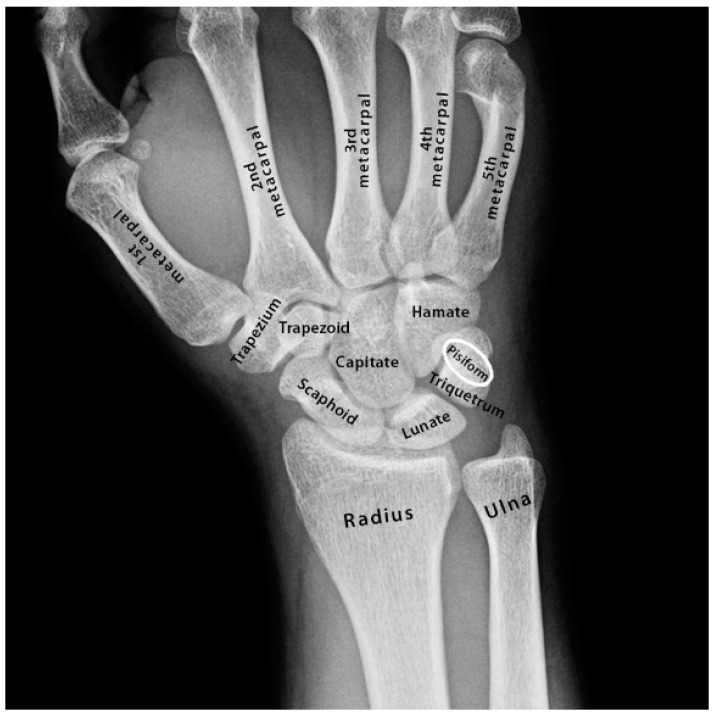
Bones of the human hand and wrist by TheSkeletalSystem.net (accessed on 10 June 2022).

**Figure 3 genes-13-01195-f003:**
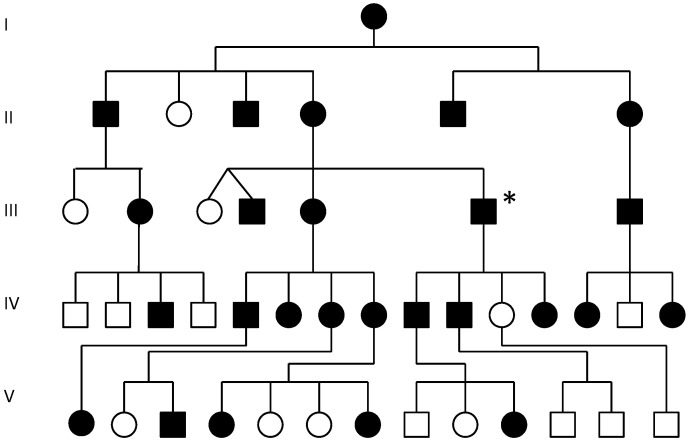
Pedigree of speech impaired family [13]: Filled symbols denote all those members of the family with evidence of SYNS4 and all of those affected that were tested had disruption of *TOSPEAK/C8ORF37AS1* [12]. All but one of the affected family members were speech impaired and all of those affected that were tested had malformation of the larynx (Figure 4B,D). The degree of speech impairment varied dramatically between affected family members. All affected family members presented with a degree of bilateral pisiform elongation. Pisiform elongations were near identical bilaterally yet varied markedly in length between affected family members (Figure 4E–K). All affected also had a degree of progressive postnatal vertebral fusion from an early age [13]. Additionally, 6/7 affected females, but none of the males, presented with age-related otosclerosis and conductive hearing impairment [13]. Near 50% of affected family members presented with congenital bilateral fusion of carpal and tarsal joints [13,14,21]. * Indicates the proband referred to in Figure 4.

**Figure 4 genes-13-01195-f004:**
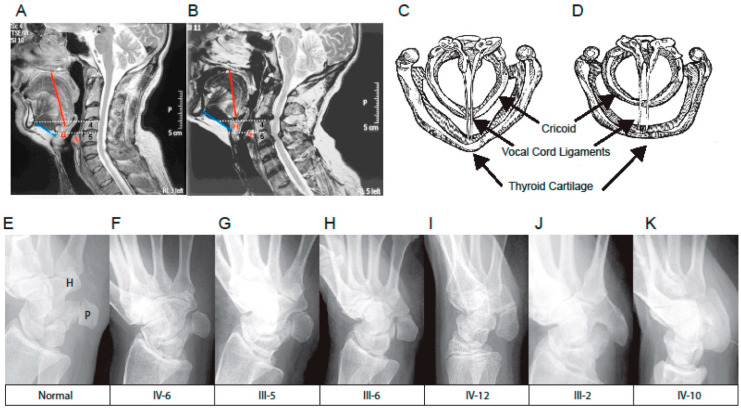
**Retrograde malformation of the larynx and wrist**. MRI analyses of the head and neck (**A**) Unaffected male from the family of matching age with a (**B**) Severely speech impaired male member of the family (III-6) with malformation of the supra-laryngeal vocal apparatus including superior (infantile/retrograde) positioning of the tongue (red line), hyoid and laryngeal cartilages—relative to the spinal axis (white dotted lines) [3,4,5,6] and increased length of the mandible (blue line). Schematic plan view of the laryngeal cartilages from endoscopic analysis for (**C**) Unaffected control and (**D**) Speech impaired male (III-6) with shortened vocal cords of altered composition and structure that failed to oppose during quiet breathing or to vibrate normally for speech or deep inspiration [14,20]. Radiological images of the wrist indicate bilateral osseous remodelling and retrograde elongation of the pisiform to varying degrees in all affected family members tested that were near identical bilaterally [8]. (**E**–**K**) Lateral images of pisiform from affected family members aligned in graded series of elongation, Arytenoid (A), Hyoid (H). Pisiform (P), Lunate (L), Ulna (U).

**Figure 5 genes-13-01195-f005:**
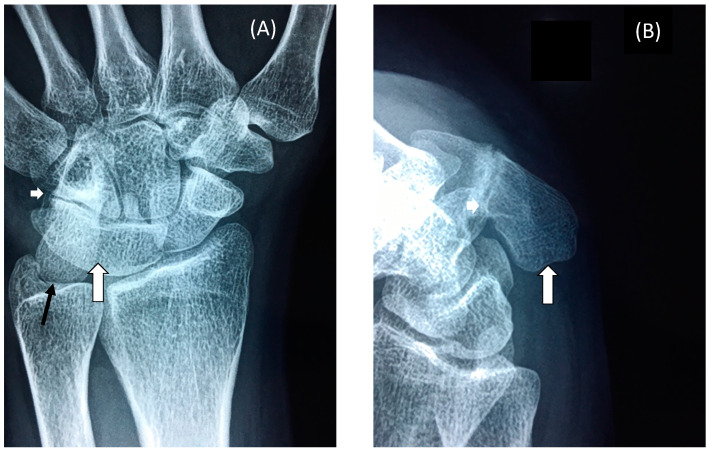
**Wrist Radiographs.** Radiographs of the wrist of an affected female member of the family (IV-12) displaying (**A**) Dorsal view of the wrist with fusion of the lunate and triquetrum (large white arrow), partial fusion of the hamate and pisiform (small white arrow) and extensive elongation of the pisiform into articulation with the radial surface of the styloid process of the ulnar (black arrow); (**B**) Side view of elongated pisiform (large white arrow) and partial fusion of hamate and pisiform (small white arrow) [13].

**Figure 6 genes-13-01195-f006:**
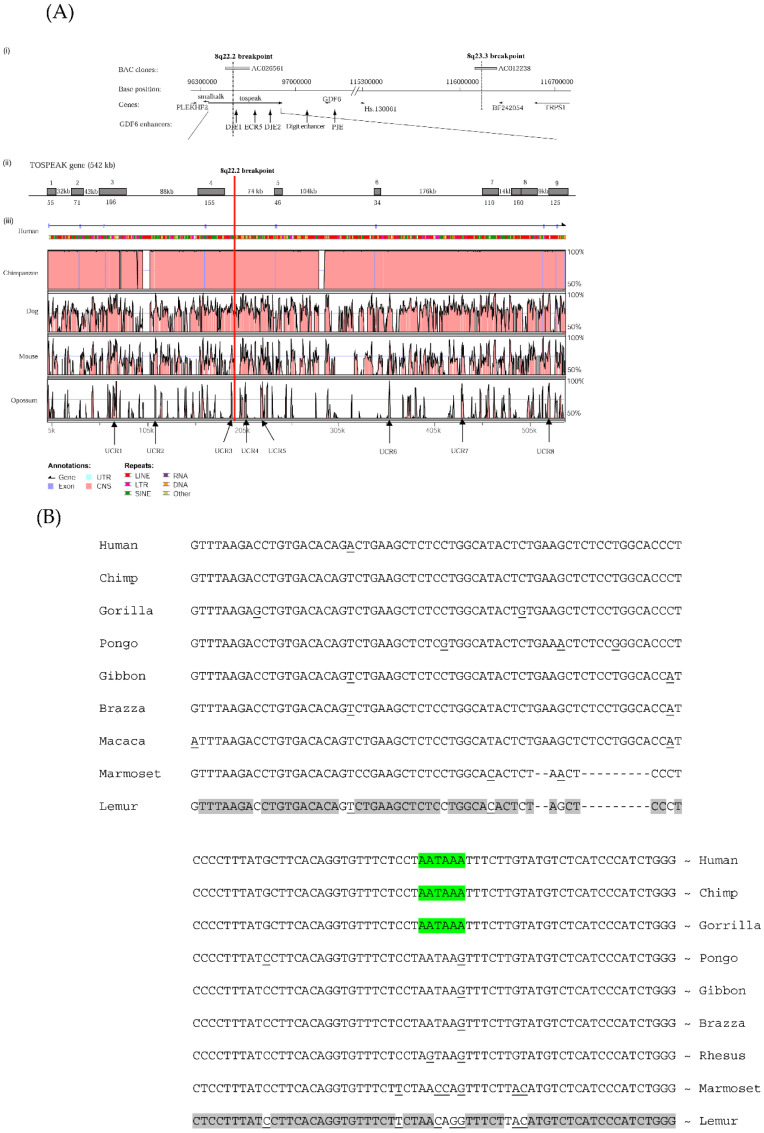
***TOSPEAK* gene characterisation** (**A**) Comparative genomic analysis of the *TOSPEAK/C8ORF37AS1[+9]* locus (**i**) Schematic view of the genomic region spanning inv(8)(q22.2q23.3) breakpoints in the speech impaired family [14]. Genes (horizontal arrows), *GDF6* enhancers (vertical arrows). (**ii**) *TOSPEAK* gene structure with 8q22.2 break point in the 4th intron. (**iii**) VISTA plot spanning *TOSPEAK* gene (exons marked blue) for multiple vertebrate species where strict selection criteria were applied for highly conserved regions (HCRs ≥ 200 bp ungapped alignment with >90% identity). The human gene annotation was obtained from the Ensembl database and the repeat information was obtained from RepeatMasker (http://www.repeatmasker.org/ (accessed on 10 June 2022). (**B**) Nucleotide sequence alignment of the last exon of *TOSPEAK* for primates with base changes underlined and conservation shaded grey; hominid polyadenylation sequence ‘AATAAA’ is shaded green while the non-hominid polyadenylation sequence is located downstream of the hominid stop site. Gaps were introduced into the sequence of Marmoset and Lemur to maximise homology.

**Figure 7 genes-13-01195-f007:**
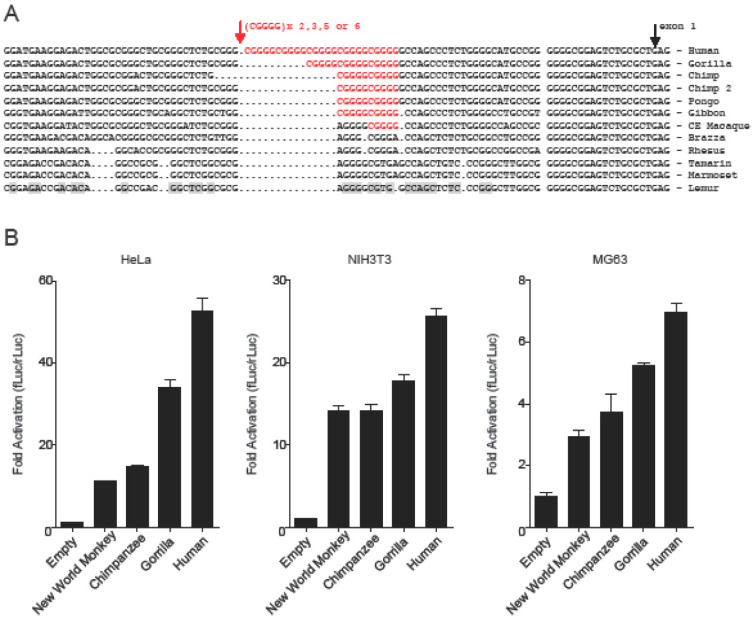
***TOSPEAK* Promoter Structure and Expression** (**A**) Nucleotide sequence alignment of *TOSPEAK* promoter. Gaps introduced to highlight maximum homology; transcription start site arrowed, site of polymorphic penta-nucleotide direct repeat cGGGG arrowed in the promoter of Gibbon, Pongo and Chimpanzee (2 direct tandem repeats), Gorilla (3 direct tandem repeats) and human (7–11 direct tandem repeats). (**B**) Promoter luciferase assays.

**Figure 8 genes-13-01195-f008:**
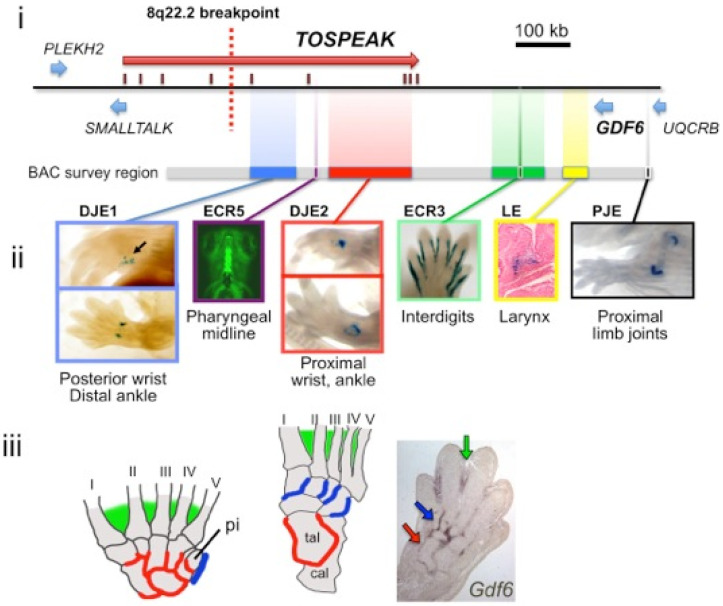
**Transgenic mouse modelling analysis of *GDF6* enhancer tissue specificity**. *GDF6* enhancer specificity mapping in transgenic mice using mouse BAC clones (Table 2) and conserved elements from transgenic reporter assays in mice or zebrafish. (**i**) The grey bar indicates the region of homology to a set of mouse BACs tested for enhancer activity (see Section 2). Coloured bars indicate positions of human alignment to murine regions that contain developmental enhancers based on BAC data (images). (**ii**) Enhancers revealed by transgenic reporters. The pharyngeal midline enhancer (ECR5) and proximal limb joint enhancer (PJE) are strongly conserved [17,23]. Note an enhancer region (blue bar) that drives expression in a small domain at the posterior side of the mouse forelimb paw along the pisiform (**ii**, left, top), and in distal hind limb joints (**ii**, left, bottom). (**iii**) Diagrams of human hand and foot skeletons, compared to *Gdf6* mRNA expression in E14.5 mouse hind limb section. Red, green and blue colour-coding in (**ii**,**iii**) reflect the skeletal domains where enhancers in (**ii**) are expressed embryonically, and also the corresponding specific domains of *Gdf6* mRNA that overlap with enhancer patterns in (**ii**). pi: pisiform. cal: calcaneus; tal: talus.

**Figure 9 genes-13-01195-f009:**
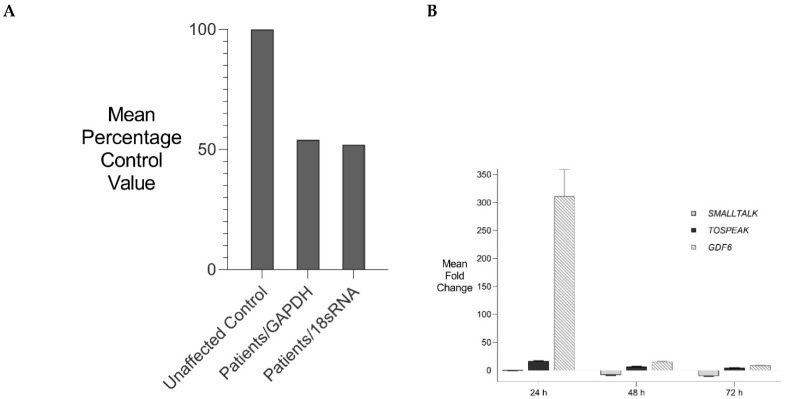
***GDF6* Expression.** Comparative rtPCR analysis of *GDF6* expression in the speech impaired family. (**A**) *GDF6* expression levels in fresh white blood cells expressed as the mean percentage change for two affected male members of the family compared with the mean for five age and gender matched unaffected controls independently normalised against the expression of two housekeeping reference genes *GAPDH* and *18sRNA* [13]. (**B**) Transient Transcriptional Interference of the *SMALLTALK*-*TOSPEAK*-*GDF6* overlapping speech gene complex: siRNA mediated knockdown of *SMALLTALK* concordant with siRNA mediated transient transcriptional interference of *SMALLTALK*. Comparative rtPCR expression analysis of *SMALLTALK*, *TOSPEAK* and *GDF6* in the normal human fibroblast cell line (NC1) over 72 h following exposure to siRNA-S1 targeting *SMALLTALK* expressed as the mean fold change relative to untreated control levels [12].

**Figure 10 genes-13-01195-f010:**
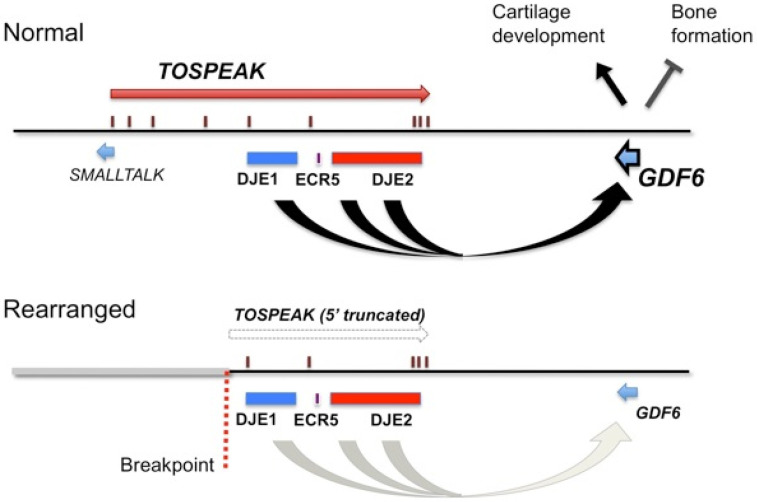
Mechanistic modelling of the *SMALLTALK-TOSPEAK-GDF6* overlapping speech gene complex before and after disruption of the *TOSPEAK* gene in the speech impaired family.

**Table 1 genes-13-01195-t001:** Variant exon structure of TOSPEAK and its transcripts in human and primates. Exons included among alternatively spliced TOSPEAK transcripts are listed for multiple primate genera including Old World Monkey (OWM) & New World Monkey (NWM). * Indicates the proband.

Human	Exon1	1A	2	3	4	5	6	7	8	8A	9	9A	GenBank
Variants	55 bp	83 bp	71 bp	163 bp	155 bp	46 bp	34 bp	110 bp	160 bp	59 bp	125 bp	95 bp	
1	*		*		*		*		*		*		GU295153
2	*				*		*		*		*		GU295154
3	*		*				*		*		*		GU295155
4	*						*		*		*		GU295156
5	*		*						*		*		GU295157
6	*						+5				*		GU295158
7	*						+5		−101		*		GU295159
8	*				*	*	*		*		*		GU295160
9					*								GU295161
10	*		*						*		*		GU295162
11	*			*			*				*		GU295163
12	*										*		GU295164
Other Primate Variants													
Chimp	*		*				*				*		GU295165
Gorilla	*		*				*		−101		*		GU295166
Orangutan	*										+185		GU295167
OWM1	28	*	*				+5		−101	*	+192		GU295168
OWM2	28	*	*				+5		−101	*	−44	*	GU295169
NWM		*					*				+192		GU295170

**Table 2 genes-13-01195-t002:** BAC Injections into mouse embryos to generate transgenic embryos.

BAC	Method	Total Embryos	Transgenics	Expressed in Interdigits	Expressed in Proximal Wrist/Ankle	Expressed in Distal Wrist/Ankle
RP23-11F19	Co-injection	71	18	7	0	0
RP23-444O1	Co-injection	46	15	0	0	0
RP23-56O6	Co-injection	84	14	0	6	0
“	Modified BAC	17	2	0	2	0
RP23-333E9	Co-injection	14	4	0	0	3
“	Modified BAC	16	2	0	0	1
RP23-354G12	Co-injection	24	2	0	0	0

## Data Availability

Not applicable.

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
