# Peer review of "Transcriptional Interference Regulates the Evolutionary Development of Speech"

_genes, 2022, doi:10.3390/genes13071195_

Round 1

Reviewer 1 Report

The author claimed primate-specific gene TOSPEAK with a human-specific promoter that associated with bone morphogenetic protein gene GDF6 with highly conserved long-range laryngeal enhancer. The relationship between the regulatory elements and the speech capacity of primates.

There are some suggestions for the manuscript:

1.     The results in Table 2 with limited numbers in positively results, the reliability of which should be evaluated. The capacity of primates' speech might relate to the brain development, the background should be showed in “Introduction”.

2.     What’s the different information among the Primates in the combined transcription factors of the duplicated “CCGGGG” sequences?   

3.     In the Figure 1, a comparative larynx structure with no capacity of speech should be added.

4.     In Figure 6A and Figure 8, the quality of the figure should be substantially enhanced. In the current figure, a lot of text are too small and the layout is not clear. The important pictures are essential for the results of the manuscript.

5.     In Figure 6B, there are two species marmoset and lemur with 2 gaps in the exon, which means the TOSPEAK genes are no function in these two primates. How do you know the homology of this gene’s evolutionary history? A phlylogenetic tree of the gene need be claimed in the ms.

6.     In  the text of Figure 7A, the sequences of several repeats in the promoter might not called “copies”.

7.     In Figure 7B and Figure 9, there are no significant testing between different control and cases.

8.   Some references should be added, from line 471 to 486 need references for this discussion.

Author Response

The author claimed primate-specific gene TOSPEAK with a human-specific promoter that associated with bone morphogenetic protein gene GDF6 with highly conserved long-range laryngeal enhancer. The relationship between the regulatory elements and the speech capacity of primates.

There are some suggestions for the manuscript:

  1. The results in Table 2 with limited numbers in positively results, the reliability of which should be evaluated. The capacity of primates' speech might relate to the brain development, the background should be showed in “Introduction”.

Response 1: Thank you for this comment

The limited number of positive results for the different BACs in Table 2 indicates high specificity for the enhancers within these BACs. This in turn provides excellent corroborative support for both the methodology and testing of the specificity of these enhancers. These tabulated results identify positive staining in multiple embryos at exactly the same anatomical sites within the different developing embryos.

  1. What’s the different information among the Primates in the combined transcription factors of the duplicated “CCGGGG” sequences?

Response 2:

This is a good comment and is best answered from the information within Figure 7 where we have included the entire sequence of the promoter on both sides of the repeating CGGGG unit for all the main primates which is as follows.

  • CE Macaque has no Sp1 transcription factor consensus binding sites
  • Gibbon has 1 Sp1 transcription factor consensus binding site
  • Pongo and Chimp have near identical promoter sequences with 1-2 overlapping Sp1 transcription factor consensus binding sites
  • Gorilla has 2-3 overlapping Sp1 transcription factor consensus binding sites
  • Human has 5-9 overlapping Sp1 transcription factor consensus binding sites
  1. In Figure 1, a comparative larynx structure with no capacity of speech should be added.

Response 3: We have included the comparison between normal and retrograde larynx in Figure 4

  1. In Figure 6A and Figure 8, the quality of the figure should be substantially enhanced. In the current figure, a lot of text are too small and the layout is not clear. The important pictures are essential for the results of the manuscript.

Response 4: I agree that the journal should try to provide the highest enhancement for these 2 figures when published as these figures do contain vital and impressive information.

  1. In Figure 6B, there are two species marmoset and lemur with 2 gaps in the exon, which means the TOSPEAK genes are no function in these two primates. How do you know the homology of this gene’s evolutionary history? A phlylogenetic tree of the gene need be claimed in the ms.

Response 5: This is a good comment.

Figure 6B presents a phylogenetic history of changes in the last exon of TOSPEAK.

We have now added the following statement into the Figure legend of Figure 6B.

“Gaps were introduced into the sequences of Marmoset and Lemur to highlight homology”

Note: As the TOSPEAK gene does not code for a protein the differences in the sequence of the last exon will not affect function per se but could conceivably alter the level of transcription.

  1. In the text of Figure 7A, the sequences of several repeats in the promoter might not called “copies”.

Response 6: The word ‘copies’ has been replaced by the words ‘direct tandem repeats’

  1. In Figure 7B and Figure 9, there are no significant testing between different control and cases.

Response 7: The different promoter constructs in Figure 7 were tested in 3 different cell lines where the

  1. Empty Vector represented a ‘no promoter control’ and where the
  2. New World Monkey represented ‘a promoter control with no repeats’.

In Figure 9 the Unaffected control PCR was compared to multiple patients using 2 different reference genes GAPDH and 18sRNA as controls

  1. Some references should be added, from line 471 to 486 need references for this discussion.

Response 8: New reference tracked in Red

Reviewer 2 Report

In this manuscript Mortlock et al., authors nicely reported evolutionary genetics of speech and its relationship with development in primates. Author presented large family group with focus on larynx and wrist development.

Author Response

In this manuscript Mortlock et al., authors nicely reported evolutionary genetics of speech and its relationship with development in primates. Author presented large family group with focus on larynx and wrist development.

Response :

We describe a large human family with a unique anatomical phenotype with speech impairment with disruption of a primate-specific gene TOSPEAK with a human-specific promoter with a laryngeal enhancer for a neighbouring bone morphogenetic protein gene. The evolution of the TOSPEAK gene and its promoter provide fascinating insights into the evolution of speech through transcriptional interference of the enhancer.